# A Simple Way to Reduce 3D Model Deformation in Smartphone Photogrammetry

**DOI:** 10.3390/s23020728

**Published:** 2023-01-09

**Authors:** Aleksandra Jasińska, Krystian Pyka, Elżbieta Pastucha, Henrik Skov Midtiby

**Affiliations:** 1Faculty of Geo-Data Science, Geodesy, and Environmental Engineering, AGH University of Science and Technology, al. Mickiewicza 30, 30-059 Cracow, Poland; 2UAS Center, The Maersk Mc-Kinney Moller Institute, University of Southern Denmark, Campusvey 55, 5230 Odense, Denmark

**Keywords:** smartphone, photogrammetry, camera calibration, 3D reconstruction, structure from motion

## Abstract

Recently, the term smartphone photogrammetry gained popularity. This suggests that photogrammetry may become a simple measurement tool by virtually every smartphone user. The research was undertaken to clarify whether it is appropriate to use the Structure from Motion—Multi Stereo View (SfM-MVS) procedure with self-calibration as it is done in Uncrewed Aerial Vehicle photogrammetry. First, the geometric stability of smartphone cameras was tested. Fourteen smartphones were calibrated on the checkerboard test field. The process was repeated multiple times. These observations were found: (1) most smartphone cameras have lower stability of the internal orientation parameters than a Digital Single-Lens Reflex (DSLR) camera, and (2) the principal distance and position of the principal point are constantly changing. Then, based on images from two selected smartphones, 3D models of a small sculpture were developed. The SfM-MVS method was used, with self-calibration and pre-calibration variants. By comparing the resultant models with the reference DSLR-created model it was shown that introducing calibration obtained in the test field instead of self-calibration improves the geometry of 3D models. In particular, deformations of local concavities and convexities decreased. In conclusion, there is real potential in smartphone photogrammetry, but it also has its limits.

## 1. Introduction

### 1.1. Motivation

The Structure from Motion (SfM) method paved the way for the use of consumer-grade cameras in photogrammetry. Drone 3D mapping has become popular, where light, small, customized cameras are used. Consumer-grade cameras, such as Digital Single-Lens Reflex camera (DSLR) or mirrorless, are successfully used in documenting monuments, museum artifacts, and geological objects, as well as in industrial photogrammetry. In recent years, smartphone cameras (SPCs) have been introduced in similar and often broader fields of applications. Consequently, a new term, smartphone photogrammetry (SPP), has been coined. It is thus worth asking whether the rules guiding the accuracy of photogrammetric measurements can be transferred to the SPP. The question is particularly important as the SPP development lacks in-depth research on the accuracy of the resulting 3D models.

### 1.2. Camera Calibration and on-the-Job Calibration in Photogrammetry

Pinhole camera calibration is a process designed to estimate interior orientation elements (IO) including principal distance (f), principal point offset (x_0_, y_0_) as well as distortion parameters [1]. The most prevalent distortion model is the Brown model [2], which includes radial (k parameters) and tangential (p parameters) distortion.

In photogrammetry surveys utilizing metric cameras, such as DMC or Vexel, the calibration is determined in a separate process. The estimated parameters are then included as fixed values in the alignment of the image bundle adjustment (BA). This strategy is based on the high stability of these cameras, which allows for reducing the BA unknowns to the elements of the images’ external orientation (EO) [3]. Semi-metric cameras (e.g., PhaseOne cameras) are also characterized by high stability of IO. Although periodic calibration is recommended [4]. Custom UAV (Uncrewed or Unmanned Aerial Vehicle) cameras’ IO stability is also improving, but a detailed assessment of this requires research over a longer period of time. Therefore, UAV cameras in photogrammetric surveys are non-metric, along with all cameras produced for general photography and not for measurement purposes.

Zhang proposed one of the most prevalent calibration methods [5]. The method uses a flat calibration field in the form of a checkerboard. It has been implemented in OpenCV and has become a de facto standard in computer vision (CV) [6,7]. The two-step procedure, where calibration is performed first and then the IO are fixed in CV operational tasks, works better the more stable the IO parameters are. A one-step method has been researched for many years, both in photogrammetry and computer vision. In this method, calibration runs simultaneously with the process of determining the EO. SfM is representative of a one-step methodology. It was created using multiple research contributions [8,9,10].

Within SfM, IO parameters are estimated as additional BA unknowns or in a separate process iteratively interlaced with BA. SfM self-calibration provides correct results providing few conditions. The tie points should be evenly distributed in the photos, and the relation between the objects’ depth range and the image acquisition distance should be large enough. However, this can be mitigated by the use of significant variability in images’ orientation angles (roll, pitch). The IO accuracy is also unfavorable when the network of images is built with only a few strips, which is typical for linear objects [11]. In those cases, the IO errors distort the EO, which leads to the so-called bowl effect [12].

The estimation of IO and EO in SfM paves the way for the generation of a dense point cloud. The process is called multi-view stereo (MVS). Two MVS solutions are used, the first is based on a sequence of stereo pairs [13] and the second on the simultaneous matching of multiple images [14]. The SfM-MVS method has been implemented in many tools, both open source (e.g., OpenCV [15], OpenDroneMap [16], Meshroom [17], MicMac [18]) and commercial (e.g., Pix4D [19], Metashape [20]).

### 1.3. Smartphone Cameras vs. DSLR

The built-in camera is a basic component of every smartphone. Additionally, more often one can find devices equipped with a set of lenses with different characteristics, e.g., a wide-angle lens, ultra-wide-angle, and telephoto lens. The size of smartphones necessitates the miniaturization of their cameras. The smartphone casing, often less than 10 mm thick, creates a small space to place the lens and matrix. In smartphones, matrices with a resolution of 12 Mpix prevail, with a frame with a 4:3 aspect ratio [21]. In DSLRs, as well as in their mirrorless counterparts, the resolution of the matrices is often 20 Mpix or slightly more. The physical sizes of the matrices differ significantly, with diagonals of the matrices in smartphones reaching 0.5″, while in DSLRs they are longer than 1″. Relatively high MPix resolution of smartphone cameras is achieved thanks to very small matrix pixel sizes, usually in the range of 1 to 2 μm. In DSLR, the matrix pixels are several times larger. A smaller pixel size of the matrix adversely affects the amount of image noise. Therefore, high-resolution SPCs images, reaching 100 Mpix, are usually heavily noisy [22]. The problem is reduced by replacing the classic Bayer filter with the so-called “multicell sensors” with pixel clusters of 4 to 9 pixels. Pixel clusters have increased sensitivity to light, resulting in a very high-resolution effect or noise reduction by pixel binning [21]. Both SPCs and DSLR cameras are multi-lens. Due to the very short space between the lens and the matrix (single millimeters), smartphones use plastic lenses instead of glass lenses found in DSLRs. Plastic lenses have a high thermal sensitivity, which causes a change in the refractive index, sharpness, and curvature of the lens field during use [21].

### 1.4. Overview of Smartphone Photogrammetry Applications

The use of smartphones in photogrammetry has been gaining popularity for some time. Thanks to the widespread use of smartphones, their computing power, and widely available easy-to-use software, more and more people decide to use smartphones for 3D modeling as an alternative to expensive, specialized equipment (DSLR cameras, laser scanners, etc.). Smartphones are mostly utilized to survey and model small objects. Literature study on smartphone photogrammetry shows most usage in the medical field, inventory of broadly understood cultural heritage, issues related to geomorphology, geotechnics, and geology, as well as strictly industry applications (e.g., displacement or volume measurement of earth masses). Table 1 lists smartphone photogrammetry publications according to the research area. The search was conducted using Scopus database, looking through research papers titles, key-word, and abstracts for key-words ‘photogrammetry’, ‘smartphone’, ‘3D’, and ‘model’. The list was then manually limited to include only relative publications.

By far the most common area where smartphones are used is the documentation of cultural heritage objects. This applies to museum artifacts [25,26], archaeological objects [23,24], and monuments (understood as single objects or complexes of objects) [27,28,29,30]. The second most popular field is medical applications. Solutions that allow recreating the shape and size of individual parts of the body [45,46,47,57,58] and models for the needs of, e.g., prostheses or plastic surgeries [48,49,51,63] are predominant here. In the case of geological issues, the analysis of rock porosity and roughness is a common topic [65,66], while in geomorphology it is popular to create digital models of various types of geomorphological forms (e.g., cliffs, caves) [70,71]. In terms of industrial applications, it is worth noting displacement determination [78,79] and utility networks modeling [75,76].

In addition to the large thematic groups described above, one can find a number of publications in which 3D models of various objects were made using images taken from a phone [80,81,82,83,84,85,86,87,88,89,90,91,92,93]. Smartphones were also used in radioactivity detection [94] or in forensic science [95,96]. It is worth mentioning that despite the increasingly frequent use of smartphone cameras for 3D modeling, it is very rare for the process to include camera calibration. The most common solution is on-the-job calibration. Among the many smartphone photogrammetry publications, only a small part of them contain information on calibration [30,40,41,59,64,67,72,74,76,78,90,93,97,98,99,100,101,102,103].

Figure 1 shows how many cited publications have been published each year. It can be seen that starting in 2016, the number of publications using smartphone photogrammetry is constantly growing and this trend will certainly continue in the future.

## 2. Materials and Methods

### 2.1. Research Aim

The main research objective was to evaluate smartphone cameras from the point of view of their suitability for photogrammetric measurements. It was assumed that the smartphone camera corresponds to the pinhole camera model. This made it possible to use analytical photogrammetry based on collinearity equations [1,3]. Before the start of the research, the experience was gained in utilizing smartphone imagery in 3D modeling within the SfM-MVS method. In these projects, the source of the IO was self-calibration, which is a common practice. However, it was noted that for the same camera calibration parameters, in particular the principal distance and the position of the principal point, would vastly vary between projects. Therefore, it was decided to investigate whether this was the result of camera IO instability, or whether the problem lies in the use of self-calibration for cases where the alignment of the bundle of images is poorly conditioned. A goal was set to develop a method minimizing the effect of IO instability of cameras on the results of a photogrammetric survey. The existing processing pipeline should be changed as little as possible, to retain usability. 

The research involved two main stages. In the first one, calibration was performed for multiple, different smartphones at specific time intervals. A checkerboard calibration method was selected. The analysis of the calibration results made it possible to separate smartphones with high and low IO stability. In the second stage, photos of a small artifact were taken with selected smartphones. Using the SfM-MVS method, 3D models in the form of a dense point cloud were developed. Two 3D models were developed for each SPC, one with self-calibration and the other with pre-calibration fixed in SfM processing. Then, shape deformations of the obtained models were analyzed based on the distance to the reference model developed with the DSLR.

### 2.2. IO Stability of Smartphone Cameras

IO stability was tested for 14 smartphone cameras from Xiaomi, Samsung, Motorola, Lenovo, Oppo, Realme, and Huawei. Only those SPCs that have the option of manual focus were selected (iPhone cameras do not have this option and were therefore omitted from this experiment). 

The calibration was based on the widely used Zhang method [5]. A chessboard displayed on a computer screen was photographed. For each calibration, at least 9 photos were taken with a wide variety of roll and pitch angles, which, while striving for the checkerboard to cover 90 to 100% of the photo frame, also forced a change in the camera-screen distance. This chessboard image acquisition strategy reduces the correlations of the estimated unknowns. The camera-calibration-with-large-chessboards program was used to calculate the calibration parameters. The internal chessboard corners were located by applying a convolution to a grayscale version of the input image with a kernel consisting of complex numbers. The structure of the kernel was chosen such that chessboard corners generate a strong response. After detection, the corner locations were enumerated, and then the OpenCV [15] camera calibration function is used to determine the camera parameters. The application is available to download under an opensource MIT license [104].

The calibration was repeated at least four times over a two-month period. The focus was always set to infinity so that for each calibration the principal distance would be identical (equal to the focal length in this case). The following were recorded from each calibration: principal distance, principal point location, radial (k1, k2, k3), and tangential (p1, p2) distortion coefficients.

### 2.3. 3D Model Deformation

The subject of modeling was the small sculpture (circa 30 × 30 × 30 cm) shown in Figure 2. Two smartphones were selected, representing different levels of IO stability. The photos were taken omnidirectionally, with the cameras stationary and the object rotating between the photos with an interval of 10°. This is a popular method for modeling small artifacts [25,26,66,69,89]. There were textured walls in the background of the sculpture, so in the development process, it was necessary to remove the so-called stationary tie points. Many researchers take photos so that the background is untextured [25,26,69]. Using the SfM-MVS method, two models in the form of a dense point cloud were developed for each SPC, one with self-calibration, and the other with pre-calibration. The SfM-MVS process was carried out using the Metashape software, which can automatically exclude stationary tie points from the SfM process [20].

Additionally, pictures of the sculpture were taken with a Nikon 5200 DSLR camera (manufactured by Nikon in Japan) in the same way as described above. The model was then developed using the SfM-MVS method with fixed pre-calibration (IO parameters were determined from the chessboard test and then considered constant). After confirming the IO stability of the Nikon D5200 camera and considering the low noise of Nikon cameras [74], the 3D model was treated as a reference. The models were compared in CloudCompare [105] using the cloud-to-cloud distance function. 

The basic characteristics of the cameras used in the research are given in Table 2. The position of each camera was selected so that the object occupied a similar part of the photo. Due to the similar horizontal angle of view of the cameras, the positions were very close to each other. The focus was set manually to a selected place on the sculpture. For each camera, a checkerboard calibration was performed, with the same focus setting as for the sculpture registration (about 1 m). The calibration was done immediately after taking the photos, while for the Samsung Galaxy S10(manufactured by Samsung in Vietnam), due to its low IO stability, the calibration was performed twice, both before and after the sculpture registration. For this SPC, the final IO parameters were calculated in one process from all photos (4 × 9).

In each SfM-MVS process, only 3 markers with known XYZ coordinates in a local object coordinate system were used. The markers were used in a rigid transformation of the model to the local coordinate system, which facilitated the comparison of the 3D models. As in the BA procedure, all ground control points take part in the adjustment, the remaining markers (visible in Figure 2) were not used to avoid their impact on local changes in the shape of 3D models.

## 3. Results

### 3.1. IO Stability of Smartphone Cameras

The analysis of the SPC’s calibration results showed significant differences between the individual camera models in terms of IO stability. The greatest variation concerns f, x_0_, and y_0_ parameters. For some cameras, the variability of these parameters was small, at the level of a few pixels, but for most of them, the variability was higher (see Figure 3). Distortion changes between calibrations of the same cameras were very small. It is true that the position of the main point affects the distortion estimation, but with high radial distortion, reaching 50 pixels at the edges of the image, this influence is small. The tangential distortion was very small for all the tested cameras, it did not exceed 2 pixels at the edges of the image.

For the tested SPCs, a stability ranking was developed, based on the volatility of f, x_0_, y_0_. Mean absolute deviation (MAD) was calculated for each of these parameters. Then, each camera was assigned three numbers from 1 to 14, resulting from the separate sorting of the three MAD statistics (from min to max). The sum of these numbers determined the position in the ranking shown in Table 3).

For comparison, the Nikon DSLR camera obtained the following MAD values of f, x_0_, y_0_: 0.31, 0.96, 0.41 pixels.

The following were selected for the next stage of the research: Xiaomi Redmi Note 11S (manufactured by Xiaomi in China) and Samsung Galaxy S10, i.e., SPC’s which ranked first and last.

### 3.2. 3D Model Deformations

Deformations of individual 3D models obtained from two SPCs were assessed by the average value of the distance (d) to the reference model and its standard deviation, the median, and the maximum distance. Table 4 summarizes these measures, the values for the variant with self-calibration and with pre-calibration are given. Pre-calibration resulted in a reduction of the average distance compared to self-calibration by 30% and 34% for Xiaomi Redmi Note 11S and Samsung Galaxy S10, respectively. The standard deviation and median also decreased. The improvement in the standard deviation is relatively small, because a small number of noise points where the distance clouds to the pattern reaches approximately a dozen mm (the Xiaomi Redmi Note 11S data is noisier).

Figure 4 shows the density plots of the distances between 3D models with SPC’s in relation to the reference model. The shape of the density plots shows that the models obtained from pre-calibration have a much higher concentration of small errors and contain much fewer large errors. For Samsung Galaxy S10, pre-calibration reduces errors in the range of 0.9–2.8 mm from 49% to 20%. For Xiaomi Redmi Note 11S there is a reduction of errors in the range of 0.7–2.4 mm from 39% to 17%.

The data in Table 4, combined with the conclusions drawn from Figure 4, clearly indicates that the models with pre-calibration are closer to the reference model than the models with self-calibration.

In order to assess the spatial distribution of errors in 3D models, visualizations showing deviations from the reference model were developed (Figure 5 and Figure 6). Views showing the sculpture from the front were selected for the presentation, as the nature of the deformation is better visible with this perspective. Figure 5a and Figure 6a show that for the variants with self-calibration, the errors are arranged in clusters around local deformations. On the other hand, deformations of models created with the use of pre-calibration are smaller in value and are more spatially dispersed (Figure 5b and Figure 6b). This phenomenon is more clearly visible in the Samsung Galaxy S10 model (Figure 5a), where the largest deformations in terms of value occur on the upper part of the sculpture (hair), while the left part of the sculpture and a fragment of the right part above the eyes show greater deformations than the right part. For Xiaomi Redmi Note, the self-calibration model has smaller deformations (Figure 6a), but the general tendency of their spatial arrangement is similar to that of the Samsung Galaxy S10. However, in the case of models with pre-calibration, a reduction of clusters with a similar distance error is visible, stronger for Xiaomi Redmi Note (Figure 6b) than for Samsung Galaxy S10 (Figure 6a).

## 4. Discussion

Calibration of various smartphone cameras showed that there are large differences in terms of IO stability. Among the examined cameras, there were those whose key IO values changed at the level of a few pixels (stable). For most, the changes were larger, and for some, they reached several dozen pixels (mainly for the principal distance). Research shows that the stability is independent of the manufacturer (e.g., Xiaomi). There is also no correlation between the market price of the smartphone and IO stability. The instability is probably due to the interference of the camera software in the final image. It is magnified by the purpose of processing, which is always the visual quality of the image, not whether the idea of a pinhole camera is met. It cannot be ruled out that the instability is influenced by lens deformations caused by temperature changes.

Regardless of the differences between smartphone models, the following observations were found: (1) in smartphones, the principal distance f and the position of the principal point x_0_, y_0_ change continuously, (2) the distortion is highly stable, and the tangential distortion is so small that it can be neglected, (3) the use of pre-calibration in the SfM-MVS method increases the accuracy of 3D modeling.

The last of the above conclusions is of great practical importance, as self-calibration, which is an optional element of the SfM-MVS method, is commonly used in SPP. This was due to the adaptation of the rules used in UAV photogrammetry. However, the cameras used in UAVs are in the vast majority much more stable than SPCs. Their production is dedicated to metric applications. In addition, the UAV captures imagery of terrain, which is beneficial for the detection of tie points throughout the entirety of the image. When taking pictures of small objects made by SPCs, such convenient situations occur less often.

Estimation of IO and EO parameters during SfM-MVS is popular, as it simplifies the process of photogrammetric 3D modeling. From a theoretical point of view, it is preferable to estimate IO and EO in separate processes. This is due to the correlation of IO and EO parameters, which increases the uncertainty of the estimation. The highest correlations occur between principal distance (f) and camera height (Z_0_) as well as the location of the main point (x_0_, y_0_) and pitch and roll camera orientation angles. The level of critical correlation coefficients can vary from a few percent to 80% [7]. The highest correlation values are obtained when the object is flat and the photos are slightly twisted relative to each other and taken from a similar distance (height). It is true that such conditions did not occur during the modeling of the sculpture, but another problem arose. The sculpture occupied only about 15% of the surface of the photos, it was always the central fragment. This causes very bad conditioning of the computational process in which EO and IO are determined. The IO-EO correlation makes the photos tie into a lattice, but both the IO and EO parameters are far from true. Continuing the MVS process on mutually agreed but actually incorrect EO and IO values, we get a deformed point cloud. Unfortunately, the SfM-MVS process does not clearly signal the situation of poorly conditioned computations.

The introduction of a separate calibration to the SfM-MVS brought positive results. 3D models for both SPCs have gained in accuracy. Global statistical measures indicate an improvement of 30%, which, however, does not objectively reflect the improvement in the geometric quality of the model. A significant benefit of the introduced strategy is the reduction of local deformations of concavities and convexities.

The improvement in model accuracy as a result of using pre-calibration was similar for both tested SPCs, with a slight advantage of the Xiaomi Redmi Note 11S over the Samsung Galaxy S10. This begs the question, why did the stable IO camera give slightly better accuracy than the unstable camera? To explain this effect, there are three things to consider: the ground sampling distance (GSD), the radiometric quality of the images, and the pre-calibration method. In the experiment, the GSD of the Xiaomi Redmi Note 11S was 17% larger than the pixel of the Samsung Galaxy S10. This caused more noise in the photos on the outline of the sculpture, where the brightness of the pixel is shaped by the rays reflected from the sculpture and from the background. The visual comparison of the images shows that the camera which is more stable in terms of IO contains more radiometric interference (visual assessment is subjective). This is also evidenced by the greater number of outliers in computed distances (relative to the average values). Probably better color demosaicing is used in the Samsung Galaxy S10 camera. On the other hand, the issue of low IO stability of the Samsung Galaxy S10 camera was suppressed by performing the calibration from four series of photos taken before and after registration of the sculpture (only one series of photos was taken for the Xiaomi Redmi Note 11S).

## 5. Conclusions

Smartphone photogrammetry is a term announced prematurely because too few studies have been conducted to verify the objectively obtained accuracy. The research has shown that IO stability varies between different SPC models. Therefore, in order to use a smartphone for metric measurements with a certain accuracy, it must be subjected to a stability test by means of independent calibrations.

Drawing conclusions from the statistics of the SfM-MVS process can be misleading and hide the deformation of the developed 3D models. We have experimentally proven that in SPP it is beneficial to perform calibration outside of SfM-MVS, in a separate process. Our solution does not restrict access to SPP, because calibration only requires taking pictures of the chessboard displayed on a computer monitor, and both open-source and commercial software can be used for calculations. In addition, we provide a program that, compared to others, is characterized by better detectability of chessboard corners.

For some applications, such as modeling of geological structures, the use of smartphone photogrammetry can be recommended without additional conditions. However, where the metric accuracy is important due to the purpose of the measurement, measures to improve the accuracy described in the article should be introduced. 

Our study did not include the influence of radiometric quality on SfM-MVS results using SPCs. We see the need for meticulous research on this aspect in order to comprehensively determine what metric potential the SPP has. We believe that in the face of many unexplained issues affecting the quality of 3D models from smartphones, you should always consider whether modeling photos should not be taken with a DSLR camera.

## Figures and Tables

**Figure 1 sensors-23-00728-f001:**
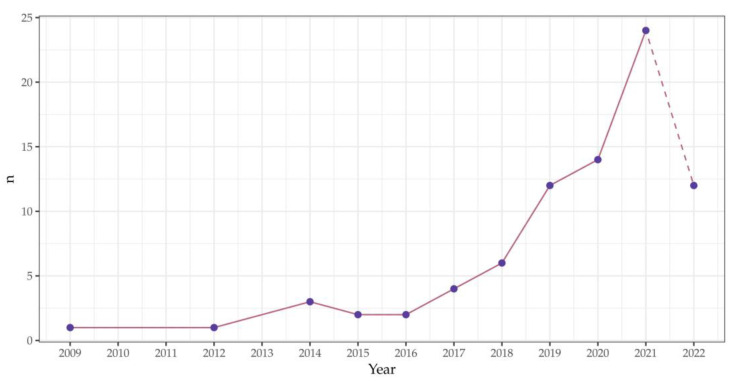
A number of publications including smartphone photogrammetry through the years. Only cited publications were used for this graph.

**Figure 2 sensors-23-00728-f002:**
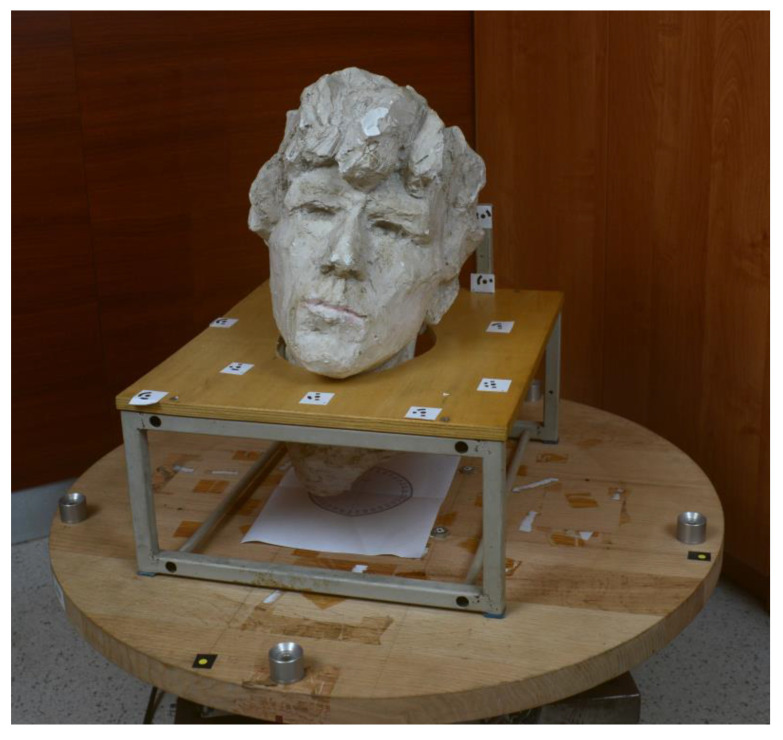
Sculpture used in the research. Ground control points are visible on the table underneath.

**Figure 3 sensors-23-00728-f003:**
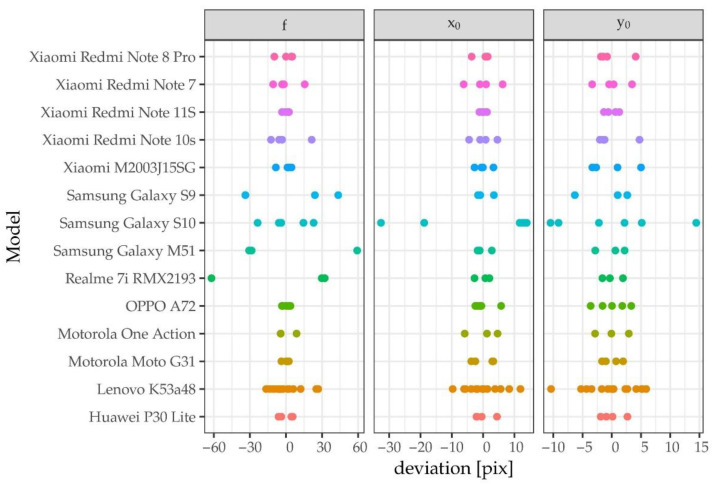
Stability of IO of all tested Models. The deviation was calculated as distance from the mean value.

**Figure 4 sensors-23-00728-f004:**
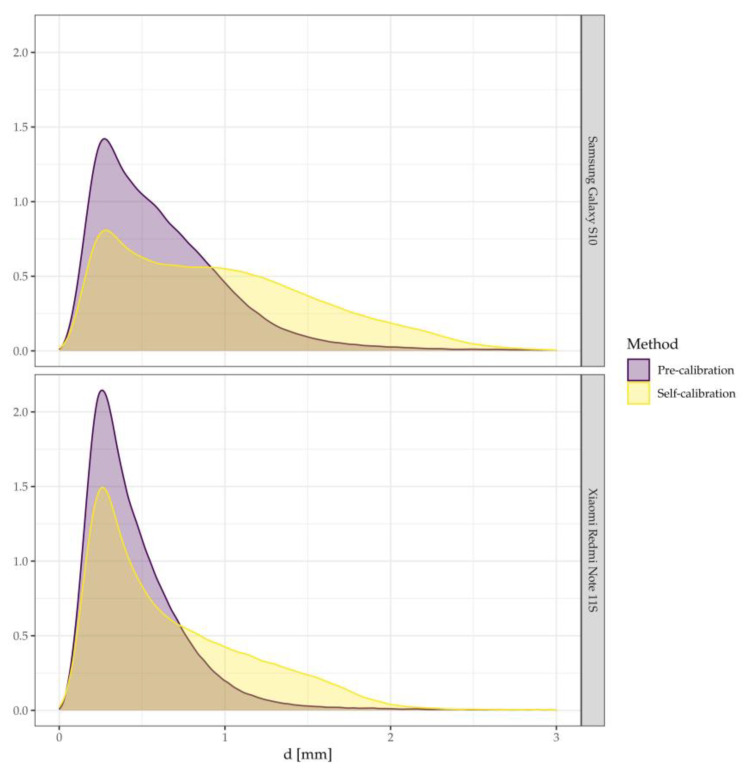
The density plot of the distances between 3D models. SPC’s in relation to the reference model.

**Figure 5 sensors-23-00728-f005:**
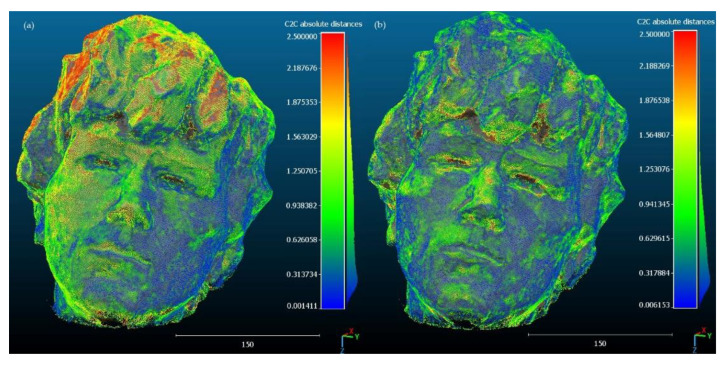
Deviations between the reference model and the model from Samsung Galaxy S10 images: (**a**) with self-calibration, (**b**) with pre-calibration.

**Figure 6 sensors-23-00728-f006:**
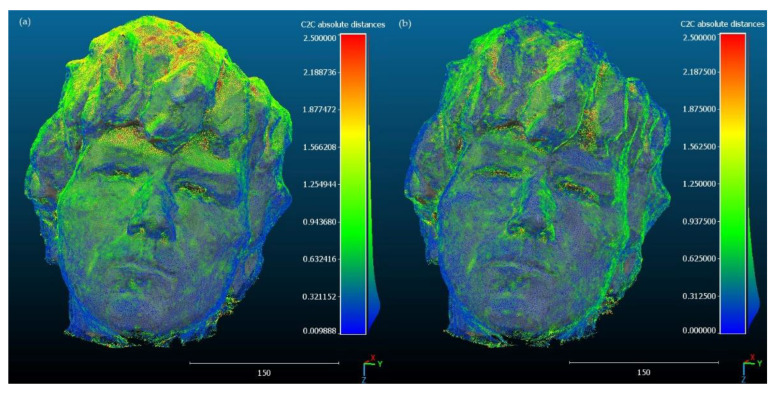
Deviations between the reference model and the model from Xiaomi Redmi Note 11S images: (**a**) with self-calibration, (**b**) with pre-calibration.

**Table 1 sensors-23-00728-t001:** Most popular research areas utilizing smartphone photogrammetry within cited papers.

Research Area	Research Papers	Number
cultural heritage	[23,24,25,26,27,28,29,30,31,32,33,34,35,36,37,38,39,40,41,42,43,44]	22
medical	[45,46,47,48,49,50,51,52,53,54,55,56,57,58,59,60,61,62,63,64]	20
geomorphology, geotechnology and geology	[65,66,67,68,69,70,71,72,73]	9
industrial application	[74,75,76,77,78,79]	6

**Table 2 sensors-23-00728-t002:** Parameters of the cameras used in the 3D model deformation studies (the principal for the focusing of the surveys).

Camera	IO Stability	Pixel Resolution H × W	Pixel Size [μm]	f [mm]	Mean GSD * [mm]
Samsung Galaxy S10	low	2268 × 4032	1.5	4.9	0.3
Xiaomi Redmi Note 11S	high	3000 × 4000	2.1	6.1	0.35
Nikon D5200	very high	4000 × 6000	4.0	21.1	0.2

* Ground sampling distance (GSD).

**Table 3 sensors-23-00728-t003:** Stability ranking for the tested SPCs.

Model	Production Year	MAD f [pix]	MAD x_0_ [pix]	MAD y_0_ [pix]	Points (f/x_0_/y_0_)	Ranking
Xiaomi Redmi Note 11S	2022	1.94	0.77	0.96	2/1/1	1
Motorola Moto G31	2021	1.91	3.11	1.33	1/10/3	2
Xiaomi M2003J15SG	2020	4.20	1.66	3.00	4/2/11	3
Huawei P30 Lite	2019	5.03	2.23	1.41	6/7/4	4
Xiaomi Redmi Note 8 Pro	2019	4.82	1.83	2.05	5/5/8	5
OPPO A72	2020	2.40	2.31	2.08	3/8/9	6
Realme 7i RMX2193	2020	41.33	1.83	1.28	14/4/2	7
Samsung Galaxy M51	2021	39.49	1.87	1.88	13/6/5	8
Xiaomi Redmi Note 7	2019	7.80	3.61	1.90	8/11/6	9
Motorola One Action	2019	5.86	3.91	1.94	7/12/7	10
Samsung Galaxy S9	2018	33.76	1.73	3.16	12/3/12	11
Xiaomi Redmi Note 10s	2021	10.69	2.70	2.37	10/9/10	12
Lenovo K53a48	2017	9.91	4.18	3.45	9/13/13	13
Samsung Galaxy S10	2019	12.51	17.15	7.26	11/14/14	14

**Table 4 sensors-23-00728-t004:** Statistical measures characterizing the errors of 3D models relative to the reference model.

SPC	Method	Mean d [mm]	Std d [mm]	Median d [mm]
Samsung Galaxy S10	Pre-calibration	0.65	0.55	0.54
Samsung Galaxy S10	Self-calibration	0.99	0.65	0.90
Xiaomi Redmi Note 11S	Pre-calibration	0.48	0.41	0.38
Xiaomi Redmi Note 11S	Self-calibration	0.70	0.54	0.54

## Data Availability

https://doi.org/10.6084/m9.figshare.21805371, accessed on 3 January 2023.

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
