# Peer review of "A Simple Way to Reduce 3D Model Deformation in Smartphone Photogrammetry"

_sensors, 2023, doi:10.3390/s23020728_

Round 1

Reviewer 1 Report

Nice work, congratuations.

A few small remarks:

1) Do not use abbreviation in the abstract, pls.

2) In table 3, add the sum of the numbers that determined the position of the smartphone in the ranking - it will be easier to analyze the data.

3) Please expand the description of Figures 5 and 6 to make it clear what they show.

4) Row 376:  "This section is not mandatory but can be added to the manuscript if the discussion is unusually long or complex." -> I think this sentence is NOT needed

Reviewer 2 Report

The authors' research objective was to evaluate the suitability of smartphone cameras for photogrammetric measurements. The study is interesting and has a certain application value. The paper states that the stability of the intrinsics of smartphone cameras is lower than that of DSLR cameras, and the principal distance and position of the principal point are constantly changing. This issue could be addressed by performing per-calibration instead of self-calibration---this improves the geometry of 3D models.

General questions:
* The authors did not tell about the settings of the SfM they were using to create 3D test models using the Metashape. From the context, one can guess that the pre-calibration variant was done using fixed intrinsics (Introduction mentions parameter fixation: 47 line "The estimated parameters are then included as fixed values in the alignment of the image bundle adjustment" and about the reference model: 203 line "The model was then developed using the SfM-MVS method with fixed pre-calibration"). Probably the test models were created the same way. So the authors used strictly two separate variants of reconstruction: the pre-calibration variant, which was based on fixed IO parameters, and the self-calibration variant, where IO parameters were initialized to some values that are not from the camera calibration. Would it be reasonable to perform another variant of reconstruction, which would be a hybrid of the first two, i.e., the pre-calibration variant with IO parameters free to adapt, or self-calibration with IO parameters initialized to the ones acquired during the calibration? What accuracy results (errors of 3D models relative to the reference model) could be expected?

* In the article, the following statements are given:
(100 line, Introduction) "Plastic lenses have a high thermal sensitivity, which causes a change in the refractive index, sharpness, and curvature of the lens field during use"
(302 line, Discussion) "The instability is probably due to the interference of the camera software in the final image."
Maybe the authors could extend the discussion regarding those two statements and the instability reasons of intrinsics. The source of instability is highly interesting.

* The authors were using Metashape (line 199) for the SfM. Could the authors give the motivation of choosing it over other programs? There is available not so small collection of open-source/commercial applications for SfM.

* The authors provided stability statistics of IO of tested smartphone cameras in Table 3 and Figure 3. The summaries could include stability evaluation results of Nikon DSLR that was used as a reference.

* It is a bit unclear how the 3D models were aligned for the comparison in CloudCompare. Were only 3 of all ground control points used or all of them? Was an iterative closest point algorithm applied to the point cloud to point cloud alignment before model comparison?

* Would the authors publish/provide the collected dataset (calibration images taken using all the cameras; images of the statue used for the 3D model) alongside the paper?

Other notes:
* Table 1 caption is the text from the template.
* The last sentence of Conclusions is the text from the template.
* Some items from the Reference (e.g., 15, 16, 17, 19, and others) need the webpage address and/or date of the last check.
* In the article, the abbreviation of the digital single-lens reflex camera is used inconsistently: DSLR and DLSR.
